# OpenReview forum: "Unspoken Hints: Accuracy Without Acknowledgement in LLM Reasoning"
_NeurIPS.cc/2025/Workshop/Reliable_ML — NeurIPS 2025 - Reliable ML Workshop_

### Official Review · Reviewer_oFc8 · 2025-09-18
**A study of the effects of diverse hinting styles on LLM Accuracy and CoT**

**Rating:** 7
**Confidence:** 3

**Review:**

The paper builds on Chen et al. 2025 to study the effect of incorrect and correct hinting on Math reasoning tasks with LLMs. They also do some preliminary experiments on hint complexity: whether the answer is presented directly as a hint or whether it's presented as a 2 or 4 operator algebraic expression.

1. The paper is well written in terms of putting enough details to make sense of the results, despite the space constraint. The limitation of MCQ-based benchmarks for the problem statement is clear, and the diversity in results across different tables conveys interesting points.
2. I would suggest that authors explore the RLHF direction more. Currently, there is a vague explanation regarding the connection between syncophany and RLHF.
3. For hint complexity, I would suggest that authors try more realistic hints. Say, for example, instead of giving the correct answer correctly or giving an expression that evaluates to the correct answer, maybe the authors can try describing how to reach the answer with a cooked-up example in the hint itself, and let the model use that example to compute its answer.

Overall, the paper is well written, and the experiments are sound for the scope of a workshop submission.

---

### Official Review · Reviewer_qkvF · 2025-09-19
**Good Experiment Setup and Interesting Takeaways**

**Rating:** 7
**Confidence:** 3

**Review:**

This paper studies the role of hints in LLM reasoning, specifically whether LLM remains faithful to their reasoning after receiving hints. This paper also specifically focuses on the cases where the hints can potentially be incorrect. The arithmetic complexity (raw answer vs equation-style) and presentation style (sycophancy vs leak-style) of the hint are also studied.

## Strengths

The experimental setup on different open-ended reasoning datasets of distinct difficulties, different hint correctness, different complexities of the hint, and different presentation styles of the hint is extensive. I do not see clear issues or missing ingredients for these setups. The results are presented and interpreted clearly. The takeaways, espectially the discovery that presentation styles matter a lot, is interesting.

## Weaknesses

As the authors mentioned, this paper adopts a very similar overall setting to *Reasoning Models Don’t Always Say What They Think* [9]. While it is true that math reasoning tasks, the possibility of incorrect hints, and the level of arithmetic complexities are all critical aspects to study, the overall setting is pretty similar to the previous work. But given this is a workshop and the overall discoveries are relevant to the comminity, I'd happy to recommend for an acceptance.